# Efficient Remediation of Cadmium- and Lead-Contaminated Water by Using Fe-Modified Date Palm Waste Biochar-Based Adsorbents

**DOI:** 10.3390/ijerph20010802

**Published:** 2023-01-01

**Authors:** Abdulaziz G. Alghamdi, Zafer Alasmary

**Affiliations:** Soil Sciences Department, College of Food and Agricultural Sciences, King Saud University, P.O. Box 2460, Riyadh 11451, Saudi Arabia

**Keywords:** magnetic biochar, sorption batch, heavy metal pollution, modeling, sorption mechanism

## Abstract

Heavy metals pollution of water resources is an emerging concern worldwide and seeks immediate attention. Date palm waste was transformed into biochar (BC), which was further modified through Fe-intercalation for the production of magnetic biochar (Fe-BC) in this study. The produced BC and Fe-BC were analyzed for chemical, proximate, surface, and elemental composition. The efficiency of the produced adsorbents to decontaminate the water from Cd^2+^ and Pb^2+^ ions was investigated through kinetics and an isotherm adsorption batch trial. Kinetics adsorption data fit well with the pseudo-second order and power function model, while equilibrium data were described well with the Langmuir and Freundlich isotherms. The maximum adsorption capacity as shown by the Langmuir model was the highest for Fe-BC for both Cd^2+^ (48.44 mg g^−1^) and Pb^2+^ (475.14 mg g^−1^), compared with that of BC (26.78 mg g^−1^ Cd^2+^ and 160.07 mg g^−1^ Pb^2+^). Both materials showed higher removal of Pb (36.34% and 99.90% on BC and Fe-BC, respectively) as compared with Cd (5.23% and 12.28% on BC and Fe-BC, respectively) from a binary solution. Overall, Fe-BC was more efficient in adsorbing both of the studied metals from contaminated water. The application of Fe-BC resulted in 89% higher adsorption of Cd^2+^ and 197% higher adsorption of Pb^2+^ from aqueous media as compared to BC. Kinetics and isotherm models as well as SEM–EDS analysis of the post-adsorption adsorbents suggested multiple adsorption mechanisms including chemisorption, pore-diffusion, and electrostatic interactions.

## 1. Introduction

The demand for fresh and clean water resources is increasing rapidly due to the prompt increase in global population [1]. It is reported that gross per-capita water availability may be reduced significantly by 2050, as the global water requirement is increasing annually [2]. In developing countries, water pollution is one of the major problems, as in these countries people do not have the resources to treat the contaminated water and use it for drinking purposes. It was reported by the World Health Organization (WHO) that about 844 million people cannot drink water which could be safe for drinking purposes, whereas 230 million of the population devote 30 min in a single day to collect drinkable water from improved sources including boreholes, piped water, rainwater, and packed water [3]. The failure in consistency in access to clean drinking water causes an increase in diseases such as diarrhea, which leads to the death of about 1.6 million people each year, and among these deaths, 90% includes children under the age of 5 years [4]. Moreover, the water quality of the resources is depleting quickly because of different kinds of inorganic and organic pollutants, which are being released from various anthropogenic activities. Moreover, natural resources are also contributing to this pollution. Human beings are being influenced by the polluted water through various approaches, including exposure to toxic chemicals which are being used to irrigate the plants. More than 700 inorganic and organic pollutants have been investigated in the water [5], and from these contaminants, heavy metals are considered to be more hazardous due to their toxic nature and persistency [6]. In the present era, industrialization and urbanization have contributed to an increase in adding heavy metals to the environment. Heavy metals contaminate the water through different activities, including mining, metal plating, tanneries, batteries, painting, and fertilizers [7]. These heavy metals are toxic and can reside in the environment for a long time as they are non-degradable. Among the most toxic heavy metals, cadmium (Cd) and lead (Pb) are of critical importance due to their wide range of applications such in fertilizer, pesticide, cosmetics, batteries, and other industries. Exposure to Pb and Cd may cause serious toxicological issues by affecting the skin, brain, pancreas, liver, and myocardium. Pb poses serious health risks to humans, such as encephalopathy, anemia, hepatitis, and nephritic syndrome [8].

Several techniques have been previously introduced by researchers for the treatment of wastewater, such as oxidation, catalytic degradation, membrane-filtration, solvent extraction, ion exchange, microbial degradation, and steam stripping [9,10,11]. However, the majority of these techniques need high electricity and more resources, which makes them very expensive to use, and therefore there is a need to develop alternative methods that should not only be cost-effective but also easily available and ready-to-use. Adsorption is a common technology worldwide as it is economic and easy. Biomass-derived biochar is considered an excellent adsorbent to adsorb the heavy metals from water and soil. Biochar (BC) is a black carbonaceous material, which is produced by biomass pyrolysis under limited-to-no oxygen supply [12]. In recent times, biochar has gained the world’s attention for various environmental and agricultural applications [13,14,15]. Due to its functional groups, high surface area, porous structure and net negative charges, BC can be used for the removal of numerous inorganic and organic pollutants [16]. However, BC cannot always perform well due to its heterogeneous structure and properties. It was reported that unmodified BC does not perform at the maximum level due to its low anti-interference capability and low adsorption capacity [17]. On the other hand, BC contains positive and negative charges at the same time; therefore, it makes the BC incapable of adsorbing anionic contaminants [18]. To overcome these limitations, scientists are paying attention to BC modification for targeted applications.

Recently, the modification of BC with foreign material via physical, biological, and chemical treatments has gained much attention as it improves the physiochemical characteristic of BC and increases its efficacy for targeted applications [19]. Additionally, different techniques have been employed by the researchers to modify BC, such as treatment with steam, gas, acids, alkali, oxidants, and microwaves [20]. For instance, Zhou et al. [21] modified BC with chitosan and observed an improvement in surface functional groups, which in turn resulted in the higher removal of Pb^2+^, Cu^2+^, and Cd^2+^ heavy metals. Similarly, Bakshi et al. [22] modified BC with hematite to fabricate zero-valent iron-BC composites and demonstrated an excellent removal capacity for arsenic. BC modification with iron has demonstrated an excellent performance for the removal of pollutants from the environment; however, it is challenging to separate BC particles from the liquid–solid phase due to their small size. Therefore, magnetic-BC composites are more conducive and efficient for contaminant removal; however, most of the methods to synthesize magnetic-BC are complicated. Moreover, there is a possibility for magnetic particles to occupy the adsorption sites of BC. Thus, new techniques are needed to design highly efficient magnetic-BC for heavy metals removal from the environment. By keeping in view the whole scenario, the innovative solution to this problem is to develop magnetic biochar composites. Magnetic biochar composites have been gaining worldwide attention in the last few decades because of their excellent adsorption capacity and magnetic separation technique [23]. Therefore, date palm tree waste was pyrolyzed to produce biochar, which was then modified to synthesize magnetic biochar in this study. The efficacy of the produced pristine and Fe-modified biochars to remove Cd^2+^ and Pb^2+^ from contaminated aqueous solution was investigated in batch-type sorption experiments.

## 2. Materials and Methods

### 2.1. Biochar Production

Local agricultural farms in Riyadh, Saudi Arabia were visited, and wastes from date palm tree pruning was collected. The collected waste was thoroughly washed with tap water followed by deionized water to get rid of any soil or dust particles, and then exposed to direct sunlight and dried in air. The petiole bases (fronds) were separated from the waste and cut into smaller pieces. Smaller-sized date palm waste (DFS) was pyrolyzed in a furnace. The DFS was placed in stainless steel boxes and placed into a furnace for pyrolysis at 550 °C for 180 min in an oven under limited oxygen supply. After the completion of pyrolysis, the box was cooled down, and the resultant biochar (BC) was weighed, grinded, and passed through a 2 mm sieve. The yield of the BC was calculated as below:(1)$$Yield\;\left(\%\right)=\frac{weight\;of\;feedstock-weight\;of\;biochar}{weight\;of\;feedstock}\times 100$$

### 2.2. Biochar Modification

The resultant BC was modified to synthetize Fe-modified biochar (Fe-BC) by using the previously reported method of Zhang et al. [24]. Specifically, a solution containing 2.92 g of Fe_2_(SO_4_)_3_ or 7.30 g of FeSO_4_7H_2_O (Merck, Darmstadt, Germany) was prepared in deionized water. In another flask, 10 g of synthesized BC was added slowly into 200 mL of deionized water under continuous stirring. The suspension was sonicated for 30 min and then stirred for 1 h for complete homogenization. The BC suspension was then added into the Fe^3+^/Fe^2+^ mixture slowly under continuous stirring, and the pH of this suspension was adjusted to 10 by the drop-wise addition of NaOH (Fisher Scientific Co., Springfield, NJ, USA). This mixture was put on a stirrer and stirred for 2 h and then allowed to sit for 24 h at ambient temperature (23 ± 2 °C). After that, the solid material was separated from the liquid–solution phase, washed several times using deionized water, dried in a vacuum oven, and labelled as Fe-BC.

### 2.3. Characterization

The produced Fe-BC and pristine BC were subjected for proximate and chemical analysis. The proximate properties were analyzed by following the standard protocols [25]. The pH was analyzed in a 1:10 ratio suspension, while cation exchange capacity (CEC) of the materials was determined through the procedure reported by Richard [26]. Functional groups were estimated by analyzing the samples on FTIR (Bruker Optics, Inc., Ettlingen, Germany), whereas the mineral composition was analyzed with the help of MAXima_X XRD-7000 (Shimadzu, Kyoto, Japan). The pore size and surface area of the adsorbents were measured with the help of TriStar II 3020, Micromeritics (USA). Surface morphology qualitative elemental composition were analyzed by a scanning electron microscope attached with EDS. The contents of hydrogen (H), carbon (C), sulfur (S), and nitrogen (N) in the samples were analyzed with the help of the CHNS-O analyzer (PerkinElmer, Waltham, MA, USA), whereas the contents of oxygen (O) were estimated by the difference method as shown in Equation (2).
O (%) = 100 − [C (%) + H (%) + N (%) + S (%) + ash (%)](2)

### 2.4. Adsorption Experiments

The efficacy of BC and Fe-BC to adsorb Cd^2+^ and Pb^2+^ from contaminated water was investigated via adsorption batch experiments. Briefly, the exact weight of Pb(NO_3_)_2_ or Cd(NO_3_)_2_·4H_2_O (LobaChemie, Mumbai, India) was dissolved in deionized water to obtain a stock solution containing 1000 mg L^−1^ of Cd^2+^ or Pb^2+^. After complete dissolution, serial dilution was carried out to prepare the metal-containing solutions with various initial concentrations of Cd^2+^ or Pb^2+^. Thereafter, the pH of the solutions was adjusted to 5.0 with the help of diluted NaOH or HNO_3_. These metal-containing solutions were then subjected to kinetics and isotherm batch experiments. Specifically, 30 mL of the metal-containing solution was taken and 1 g L^−1^ adsorbent was added. This mixture was put on the shaker for a specific period of time (0–1440 min) at 150 rpm. All the trials were replicated three times, and control treatments (without any adsorbent) were also included. After a specific time interval, the solution was filtered with Whatman-42 filter papers and the left-over concentrations of Cd^2+^ or Pb^2+^ in the solutions were measured with the help of an ICP-OES (PerkinElmer Optima 4300 DV, USA).

The amount of Cd^2+^ or Pb^2+^ adsorbed on the adsorbent can be calculated by following Equation (3):(3)$$Q_{t}=\frac{V}{m}\;\left(C_{0}-C_{t}\right)$$
where *Q_t_* is the amount of adsorbate (mg g^−1^) at time *t*, *C*_0_ and *C_t_* are initial concentration and remaining adsorbate after the adsorption process (mg L^−1^), respectively, *V* represents volume (L), and *m* represents the mass of the adsorbent (g).

Kinetics adsorption trials were conducted by using the initial concentration of 100 mg L^−1^ of Cd^2+^ or Pb^2+^, and an initial pH of 5.0. At pH 5, Cd^2+^ or Pb^2+^ ions are mobile and soluble in aqueous solution. Different time intervals such as 0, 15, 30, 60, 120, 180, 300, 480 and 720 and 140 min were used in the kinetics experiments. To perform the isotherm adsorption experiments, various initial concentrations of Pb^2+^ (0, 5, 10, 25, 50, 100, 200, 400, and 600 mg L^−1^) and Cd^2+^ (0, 5, 10, 30, 60, and 100 mg L^−1^) with an initial pH of 5.0 were used. The equilibrium studies were conducted for 1440 min of equilibrium time.

The regeneration trials of Fe-BC and BC were conducted using 0.2 M HCl up to 4 cycles. The initial concentrations for Cd^2+^ and Pb^2+^ in regeneration studies were 100 and 600 mg L^−1^, respectively. Briefly, Cd^2+^- or Pb^2+^-loaded adsorbents were washed with deionized water to remove surplus metal, and dried in an oven at 60 °C. Thereafter, the dried adsorbents loaded with Cd^2+^ or Pb^2+^ were suspended into 30 mL of 0.2 M HCl solution and shaken for 1440 min at 150 rpm. Then, the solutions were separated from the adsorbent and subjected to Cd^2+^ and Pb^2+^ analyses by ICP-OES, while the adsorbents were washed with deionized water, dried, and subjected to another cycle of adsorption–desorption.

Different kinetics as well as isotherm models were applied to investigate the adsorption process of Cd^2+^ or Pb^2+^ by BC and Fe-BC.

The pseudo-first-order kinetic model was applied as shown in Equation (4).
(4)$$ln\left(q_{e}-q_{t}\right)=lnq_{e}-k_{1}^{\prime }t$$
where *k’*_1_ and *q_e_* stand for rate constants and adsorption capacity (mg g^−1^), respectively.

The pseudo-second order was applied as shown in Equation (5).
(5)$$\frac{t}{Q_{t}}=\frac{1}{k_{2}q_{e}^{2}}+\frac{t}{q_{e}}$$
where *k*_2_ are the pseudo-second-order rate constants.

Elovich kinetics were applied as shown in Equation (6).
(6)$$Q_{t}=\frac{1}{\beta }\;Ln\alpha \beta +\frac{1}{\beta }\;Lnt$$
where *α* stands for initial adsorption in mg g^−1^ min^−1^, and *β* represents the Elovich rate constant.

The expression of the intraparticle diffusion kinetics model is shown in Equation (7).
(7)$$Q_{t}=c+k_{id}t^{0.5}$$
where *c* represents the intraparticle diffusion constant and *k_id_* stands for the apparent diffusion rate constant [(mg g^−1^)^0.5^].

The non-linear form of the Freundlich isotherm is shown in Equation (8).
(8)$$q_{e}=K_{F}C_{e}^{1/n}$$
where *n* and *K_F_* represent adsorption intensity and the Freundlich isotherm constant (mg g^−1^) (L mg^−1^)^n^, respectively.

The non-linear form of the Langmuir isotherm is expressed in Equation (9).
(9)$$q_{e}=\frac{Q_{L}C_{e}K_{L}}{1+K_{L}C_{e}}$$
where *Q_L_* is the maximum adsorption capacity (mg g^−1^), and *K_L_* is the Langmuir isotherm constant (L mg^−1^).

## 3. Results and Discussion

### 3.1. This Adsorbent Characterization

Some of the chemical and proximate analyses of DFS, BC, and Fe-BC are shown in Table 1. The results found that the pH of the DFS was noted less in comparison to BC and Fe-BC. After pyrolyzing the material, the pH was increased to 9.26 in BC from 6.78 in DFS. The exclusion of acidic groups and condensation of alkali salts could have resulted in pH increment [27]. DFS showed the lowest CEC (54.78 cmol kg^−1^), which was increased by 15% (63.19 cmol kg^−1^) in BC and up to 87% in Fe-BC (81.74 cmol kg^−1^). The highest CEC of Fe-BC than that of BC could be due to the presence of Fe-particles. Results showed that Fe contents in BC were 0.12%, while in Fe-BC they were 19.05%, respectively. The produced BC showed 35.47% of the yield and 1.02% moisture contents, whereas DFS and Fe-BC showed 3.21% and 2.08% of moisture contents, respectively. DFS showed the highest volatile matter (62.88%), which reduced almost 4-fold in BC and 4.7-fold in Fe-BC. On the contrary, the ash contents increased from 15.45% in DFS to 25.98% in BC and 51.03% in Fe-BC. Likewise, the resident matter was increased with the pyrolysis process from 18.51% in DFS to 56.79% in BC and 33.49% in Fe-BC. However, the lower resident matter contents of Fe-BC than BC could be due to the presence of Fe-particles in Fe-BC, which might have been confused with ash contents [18]. Higher ash contents in BC materials as compared to DFS could be due to the formation of some mineral compounds due to the thermalization process [28].

Table 2 presents the surface area and elemental contents of the produced adsorbents. It was noted that the C contents were enhanced from 41.82% in DFS to 68.82% in BC and 72.98% in Fe-BC. In contrast, O, N, and H were decreased with the pyrolysis process due to dehydration and depolymerization [29]. However, H contents in Fe-BC were higher than pristine BC, which could be due to H-bonding of Fe with the biochar surface. Fe-BC represented the highest Brunauer–Emmett–Teller (BET) surface area (119.82 m^2^ g^−1^), while BC and DFS showed the lowest (68.49 and 2.28 m^2^ g^−1^, respectively). Likewise, Fe-BC showed the lowest pore size (4.72 nm) as compared to BC (9.18 nm) and DFS (31.71 nm). The highest surface area of Fe-BC may be because of the existence of Fe-particles, while the lower pore size could be due to aromaticity and the loss of volatile matter (as shown in Table 1) [27].

The functional groups’ composition was observed with the help of an FTIR, and the results are shown in Figure 1. A band representing O–H stretching was observed between 3300–3500 cm^−1^ in DFS, which was removed with pyrolysis in BC, while it appeared again in Fe-BC spectra. This band was probably associated with moisture contents in the sample [18,27]. A band with large intensity was seen at 1050–1100 cm^−1^ in DFS and BC, while its intensity was significantly reduced in Fe-BC. This band was due to the existence of C–O–C stretching of polysaccharide cellulose [27]. A small band at 3450 cm^−1^ in Fe-BC was proof for the presence of Fe-particles (ferrioxyhydroxide (FeOOH)) in this composite material as these bands represent OH stretching vibrations [30]. Similarly, a band around 874 cm^−1^ in Fe-BC could be due to the presence of Fe particles. Likewise, the mineralogical composition of the produced amendments as assessed by XRD analysis is shown in Figure 2. The presence of mellite was witnessed in DFS at 21.48° 2θ, but it was removed during pyrolysis and could not be seen in BC and Fe-BC. The same happened with calcite mineral (29.21° 2θ; JCPDS: 47-1743), which was substantially reduced in BC and Fe-BC as compared to DFS. Akin to FTIR, the presence of the Fe particles was also witnessed by the XRD results of Fe-BC by observing a peak at 44.21 and 63.25° 2θ (JCPDS 00-001-1267). Likewise, a peak representing hematite (JCPDS Card No. 01-073-0603) was found at 35.54° 2θ in the Fe-BC material. These results represent significant variations in chemical, proximate, morphological, and structural properties of DFS, and the produced BC and Fe-BC. Moreover, the successful enrichment of Fe into biochar was also noticed.

### 3.2. Kinetic Adsorption of Cd^2+^ or Pb^2+^

The performance of the produced BC and Fe-modified BC to adsorb Cd^2+^ or Pb^2+^ from contaminated water was assessed in batch-type adsorption experiments. The effect of contact time on the adsorption of Cd^2+^ or Pb^2+^ onto the adsorbents is shown in Figure 3. The adsorption of Cd^2+^ or Pb^2+^ followed three phases, i.e., rapid, relatively slower, and equilibrium. A relatively quicker adsorption for both of the metals was seen during the first 200 min due to the accessibility of plentiful adsorption sites, followed by a relatively slow adsorption and eventually equilibrium stage. Overall, the adsorption of Pb^2+^ was higher than Cd^2+^ onto both of the adsorbents. This is because Pb^2+^ contains a negative logarithm of hydrolysis constant (7.71), which is less than that of Cd^2+^ (10.1) and helps Pb^2+^ to be adsorbed easily via surface complexation [31]. The Fe-BC exhibited a higher adsorption capacity compared with the pristine BC.

The adsorption data were subjected to different kinetics models. The kinetics parameters and coefficient of correlation (R^2^) are presented in Table 3. The results showed that the pseudo-second order fit well with the adsorption data for both of the studied metals (R^2^ = 0.93–0.98), followed by power function (R^2^ = 0.85–0.98). However, other models were marginally fit to the adsorption data (R^2^ = 0.93–0.98). The initial sorption rates (*h*) as presented by the pseudo-second order were highest for Fe-BC for the adsorption of Cd^2+^ (0.39 mg g^−1^ m^−1^) and Pb^2+^ (0.52 mg g^−1^ m^−1^), while pristine BC showed the lowest h for both Cd^2+^ and Pb^2+^ (0.23 and 0.08 mg g^−1^ m^−1^, respectively). Moreover, the pseudo-second order presented that the highest adsorption capacity was higher for Fe-BC both for the adsorption of Pb^2+^ (105.49 mg g^−1^) and Cd^2+^ (54.49 mg g^−1^) in comparison to the maximum adsorption capacity of BC for Pb^2+^ (45.89 mg g^−1^) and Cd^2+^ (27.12 mg g^−1^).

The best fitness of the pseudo-second order model for Pb^2+^ and Cd^2+^ adsorption on the Fe-BC and BC suggests the occurrence of the chemisorption mechanism (Reddy and Lee, 2014). Followed by the pseudo-second order model, the power function model presented the R^2^ values 0.98–0.97 for Cd^2+^ adsorption and 0.85–0.95 for Pb^2+^ adsorption. The rate coefficient (*k_f_)* values of the power function model were higher for Fe-BC (0.70 mg g^−1^ m^−1^) and BC (0.52 mg g^−1^ m^−1^) for Pb^2+^ adsorption as compared with Cd^2+^ adsorption (0.58 and 0.46 mg g^−1^ m^−1^ for Fe-BC and BC, respectively). The rate constants (*b*) of the power function model were found as 0.13 for both the adsorbents for Cd^2+^ adsorption, while it was found as 0.10 and 0.02 for BC and Fe-BC, respectively, for Pb^2+^ sorption. The suitability of the power function model proposed the occurrence of homogenous chemisorption adsorption of Cd^2+^ and Pb^2+^ onto the produced adsorbents, which is also in agreement with fitness of the pseudo-second order [32]. Overall, the rate coefficients of the kinetics models were higher for Fe-BC than pristine BC, suggesting better adsorption of these metals onto Fe-BC. Moreover, the marginal fitness of the intraparticle diffusion model advocated that pore diffusion has also aided the adsorption of Cd^2+^ and Pb^2+^ onto BC and Fe-BC. Overall, Fe-BC was found to be a more efficient adsorbent for the removal of Cd^2+^ and Pb^2+^ from the contaminated aqueous media.

### 3.3. Equilibrium Adsorption of Cd^2+^ or Pb^2+^

Isotherm models including Freundlich and Langmuir were applied on Pb^2+^ and Cd^2+^ ions adsorption data, and the constructed isotherms are shown in Figure 4. From the applied isotherm models, initially a quicker adsorption was observed with lower initial concentrations of Pb^2+^ and Cd^2+^ ions, indicating the presence of abundant free active sites for adsorption, subsequently making L-type isotherm curves. However, at the later stage, with increasing the initial concentrations of Pb^2+^ and Cd^2+^ ions, the adsorption sites were occupied with Pb^2+^ and Cd^2+^ ions and the curves moved to the right, showing a slower adsorption and ultimately leading to the equilibrium stage. Overall, the adsorption capacity of Fe-BC was far greater compared with BC in adsorbing both of the studied metals. On the other hand, both of the adsorbents showed a higher affinity towards Pb^2+^ adsorption, and removed almost all of the Pb^2+^ at 100 mg L^−1^. Therefore, the produced adsorbents remained more efficient in removing Pb^2+^ compared with Cd^2+^ ions from contaminated water.

The parameters presented by the isotherm curve fittings are presented in Table 4. Both of the applied models fit best with the Pb^2+^ adsorption, whereas Cd^2+^ adsorption data were only fit best with the Langmuir isotherm as is indicated by the R^2^. The maximum adsorption capacity estimated by the Langmuir model (Q*_L_*) was highest for Fe-BC for adsorbing both of the metals. Overall, the maximum adsorption capacities for Pb^2+^ were higher than for Cd^2+^. Q_L_ for Pb^2+^ adsorption onto BC and Fe-BC was 160.07 and 475.14 mg g^−1^, respectively, while it was 26.78 and 48.44 mg g^−1^ for Cd^2+^ adsorption, respectively. Likewise, the Freundlich isotherm predicted K_F_ values were higher for Fe-BC for the adsorption of both metals (23.30 and 129.06 L g^−1^ for Cd^2+^ and Pb^2+^, respectively) as compared to the pristine BC (10.56 and 36.66 L g^−1^, for Cd^2+^ and Pb^2+^, respectively). The 1/*n* values presented by the Freundlich isotherm for Pb^2+^ adsorption were found as 0.207 for Fe-BC and 0.295 for BC, while these values were 0.345 for BC and 0.031 for Fe-BC in the case of Cd^2+^ adsorption. These results indicated that 1/*n* values for both of adsorbents for Cd^2+^ and Pb^2+^ adsorption were less than 1, suggesting that the adsorption was satisfactory at initial lower concentration because of surface loading. Moreover, 1/*n* values for Fe-BC were lower than pristine BC for both of the models, suggesting more favorable adsorption onto Fe-BC than unmodified BC [18]. The fitness of both isotherm models indicated the involvement of both monolayer and multilayer adsorption mechanisms. Overall, the higher adsorption of Pb^2+^ onto both of the adsorbents could partially be due to diffusion into the pores as well [33]. The high surface area and pore structure of Fe-BC could have adsorbed Cd^2+^ and Pb^2+^ by pore diffusion and physical adsorption [34]. Moreover, ion exchange and electrostatic interactions can also affect the adsorption of metals onto the BC surface [35,36]. The higher adsorption capacity of Fe-BC for both of the studied metals could be due to the presence of Fe_2_O_3_/Fe_3_O_4_ in this material, which were uniformly distributed on the surface of the adsorbent. Therefore, the adsorption of metals by Fe-BC could be due to a collective effect of complexation, electrostatic attraction, diffusion, and ion exchange, which supported the maximum adsorption of metals, whereas this phenomenon was not possible with pristine BC. Moreover, regeneration studies showed that Cd^2+^ and Pb^2+^ removal efficiencies of Fe-BC were 29.58% and 45.31% after 4 cycles, respectively, which was lower than that of BC (38.13% and 53.38%, respectively). Reduction in Fe-BC efficiency could be due to oxidation and dissociation of Fe particles [18]. The maximum adsorption capacities of different magnetic BCs for Cd^2+^ and Pb^2+^ have been compared and enlisted in Table 5. It can be seen that the maximum adsorption capacity of Fe-BC for Cd^2+^ and Pb^2+^ in current study was much higher than various previously tested magnetic BCs in the literature. Hence, these results suggested that Fe-BC was an effective, robust, and sustainable adsorbent for removing Cd^2+^ and Pb^2+^ from the contaminated water. 

### 3.4. Adsorption Mechanism

The impacts of the coexisting metals ions (Pb^2+^ and Cd^2+^) on adsorption capacity are shown in Figure 5a. The binary adsorption system affected the adsorption of both the coexisting metals, and demonstrated the competitive effect of Pb^2+^ and Cd^2+^ on the removal of each other. As presented in Figure 5a, Pb^2+^ adsorption was far greater than Cd^2+^ on both Fe-BC and BC. The pristine BC could remove 5.23% of Cd^2+^ and36.34% of Pb^2+^ from a binary solution containing 100 mg L^−1^ of both of aforementioned metals. Similarly, Fe-BC proved to be an excellent adsorbent by removing 99.90% of Pb^2+^ and only 12.28% of Cd^2+^ from the binary solution. These results suggest more adsorption of Pb^2+^ as compared to Cd^2+^. It could be stated that a complexation interaction may have occurred between the metals in the binary system. Multiple mechanisms were involved in the sorption of Pb^2+^ and Cd^2+^ on the Fe-BC and pristine BC, including electrostatic interaction, ion exchange, surface complexation, chemical sorption, and intraparticle diffusion [34]. The higher adsorption of Pb^2+^ could be due to its bigger ionic size (Cd^2+^ = 1.03 Å and Pb^2+^ = 1.32 Å), which may create the Pb^2+^ aquo-cation, ultimately attracting the adsorption sites in vicinity and stabilizing the adsorbate [42]. Moreover, the hydrated radius (4.01 Å) and negative log of hydrolysis (7.71) of Pb^2+^ is less than that of Cd^2+^ (4.26 Å and 10.1, respectively) which might have favored its adsorption to the adsorbent [43]. Hence, Pb^2+^ exhibited the competitive adsorption advantage over Cd^2+^.

The solution’s pH is one of the critical factors influencing the adsorption process and competition of metals ions for the adsorption sites. The net surface charge as well as the type of speciation of the metal is highly dependent on the solution pH. It has been reported previously that metals such as Pb^2+^ and Cd^2+^ exist in ionic form below pH 7.0, and thus, can easily be adsorbed on the adsorbent through electrostatic interactions and binding on surface functional groups [44]. Generally, at lower pH levels, H^+^ is bounded with functional groups present on the surface of the adsorbent, subsequently attracting metals ions for adsorption. A further increase in pH may result in deprotonation, consequently making surface functional groups adsorption sites for metal ions [45]. In the current study, the pH of the solution for both BC and Fe-BC remained below 7.0 (initial pH 5 was used) as shown in Figure 6. Further, the pH in the Fe-BC-metal suspensions was lower than BC-metal suspension. Therefore, the lower pH levels of the adsorbent–adsorbate suspension suggested the adsorption chemical adsorption electrostatic interactions were involved in the adsorption of Pb^2+^ and Cd^2+^ on BC and Fe-BC. These results are in agreement with the results of isotherm and kinetic model simulations. The kinetic adsorption of Pb^2+^ and Cd^2+^ suggested the involvement of chemisorption and pore diffusion of metals, while the results of isotherm models suggested the presence of both the monolayer and multilayer adsorption of metals on the heterogeneous surface of the adsorbents. The adsorption of Pb^2+^ and Cd^2+^ on the produced adsorbents was further confirmed by SEM–EDS analysis (Figure 7). The pristine BC has 0.19% of Fe contents, which were increased to 32.27% within magnetic BC (Fe-BC). The EDS results revealed that the pristine BC and Fe-BC have no Pb^2+^ or Cd^2+^. However, after the adsorption of Pb^2+^ and Cd^2+^ from the contaminated water, BC showed 3.59% of Pb^2+^ and 4.59% of Cd^2+^, while Fe-BC showed 5.24% of Pb^2+^ and 5.66% of Cd^2+^. The presence of Pb^2+^ and Cd^2+^ in higher amounts after the adsorption trials is indication for the successful adsorption of these metals on BC and Fe-BC. Further, the higher adsorption of metals onto Fe-BC than that of BC is obvious with the higher weight percentages of Pb^2+^ and Cd^2+^ in Fe-BC after adsorption trials. The reduction in Fe contents from 32.27% in Fe-BC to 27.43% in Cd^2+^-loaded Fe-BC and 10.12% in Pb^2+^-loaded Fe-BC suggested the leaching of Fe-particles during the adsorption trials, which might have resulted in reducing the adsorption capacity of these adsorbents after four cycles of regeneration compared with BC. On the contrary, Fe-BC showed a higher recovery under applied magnetic field compared with BC (Figure 5b), which suggests the higher potential of Fe-BC for regeneration and repeated application. However, even with the reduced activity of Fe-BC after four regenerations, its Pb^2+^ and Cd^2+^ removal efficiency was relatively higher than most of the previously reported magnetic biochars. Therefore, modification of BC with Fe could be used as an excellent adsorbent for the removal of Pb^2+^ and Cd^2+^ from contaminated water on a sustainable basis.

## 4. Conclusions

Date palm waste was pyrolyzed to produce biochar (BC), which was subsequently modified to produce Fe-modified biochar (Fe-BC). The synthesized adsorbents were analyzed for chemical, structural, and surface properties, and successfully employed for the adsorption of Cd^2+^ and Pb^2+^ from the contaminated water. The results of the modeling data showed the best suitability of the adsorption data to the pseudo-second order and power function kinetic models, as well as to the Freundlich and Langmuir isotherm models. The best fitness of the adsorption date to the Freundlich and Langmuir isotherm models suggested the occurrence of mono- and multilayer adsorption. The adsorption of Pb^2+^ was seven to eight-fold higher than the adsorption of Cd^2+^ on both of the used adsorbents. Overall, Fe-BC was more efficient in removing both the Cd^2+^ and Pb^2+^ ions from the contaminated solution. Fe-BC application adsorbed 89% more Cd^2+^ and 197% more Pb^2+^ in comparison to BC, indicating the higher efficacy of Fe-BC towards both Cd^2+^ and Pb^2+^ ions. Cd^2+^- and Pb^2+^-loaded adsorbents were analyzed with SEM–EDS. The presence of Cd^2+^ and Pb^2+^ on the adsorbents was observed in post-adsorption SEM–EDS results, suggesting successful adsorbate–adsorbent bindings. Hence, Cd^2+^- and Pb^2+^-contaminated wastewater can effectively be remediated with Fe-modified BC on a sustainable basis.

## Figures and Tables

**Figure 1 ijerph-20-00802-f001:**
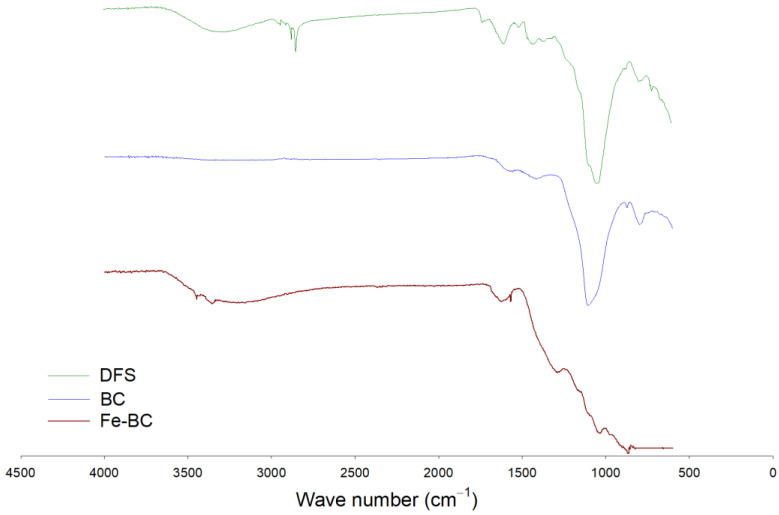
FTIR analysis of date palm waste (DFS), date palm waste biochar (BC), and Fe-modified biochar (Fe-BC).

**Figure 2 ijerph-20-00802-f002:**
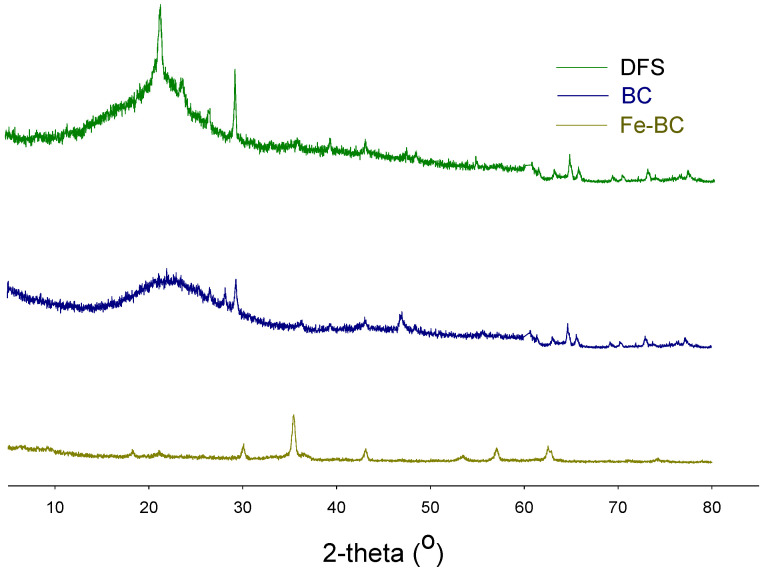
X-ray diffraction analysis of date palm waste (DFS), date palm waste biochar (BC), and Fe-modified biochar (Fe-BC).

**Figure 3 ijerph-20-00802-f003:**
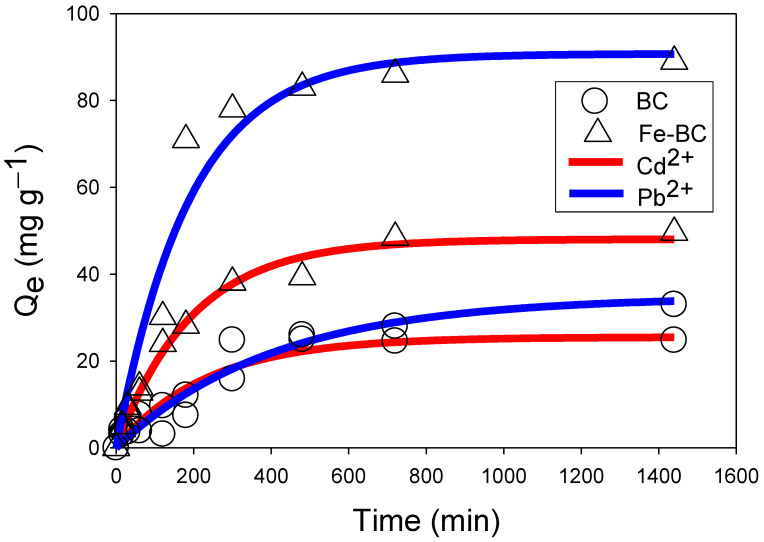
Effect of contact time on the adsorption of cadmium (Cd^2+^) and lead (Pb^2+^) on date palm waste biochar (BC) and Fe-modified biochar (Fe-BC).

**Figure 4 ijerph-20-00802-f004:**
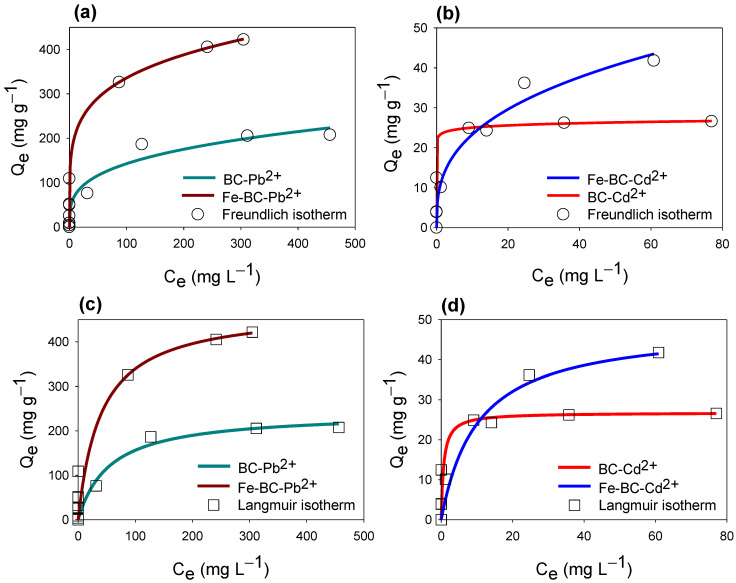
Isotherm curves for Freundlich (**a**,**b**) and Langmuir (**c**,**d**) models’ fittings for cadmium (Cd^2+^) and lead (Pb^2+^) adsorption on date palm waste biochar (BC) and Fe-modified biochar (Fe-BC).

**Figure 5 ijerph-20-00802-f005:**
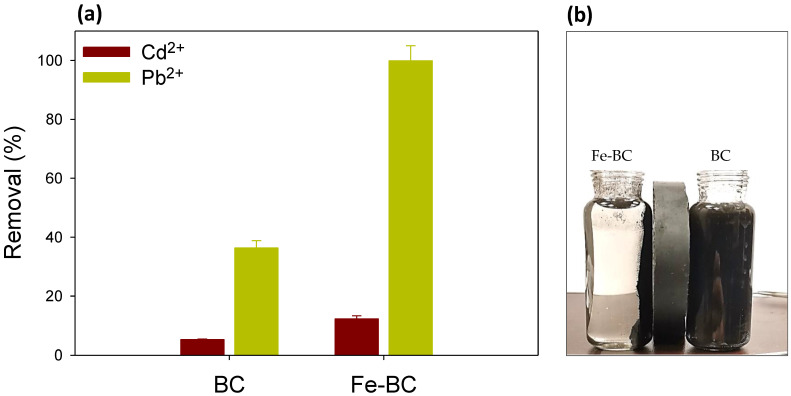
(**a**) Competitive removal of cadmium (Cd^2+^) and lead (Pb^2+^) ions from binary solution on date palm waste biochar (BC) and Fe-modified biochar (Fe-BC), (**b**) attraction of BC and Fe-BC materials to the applied magnetic force.

**Figure 6 ijerph-20-00802-f006:**
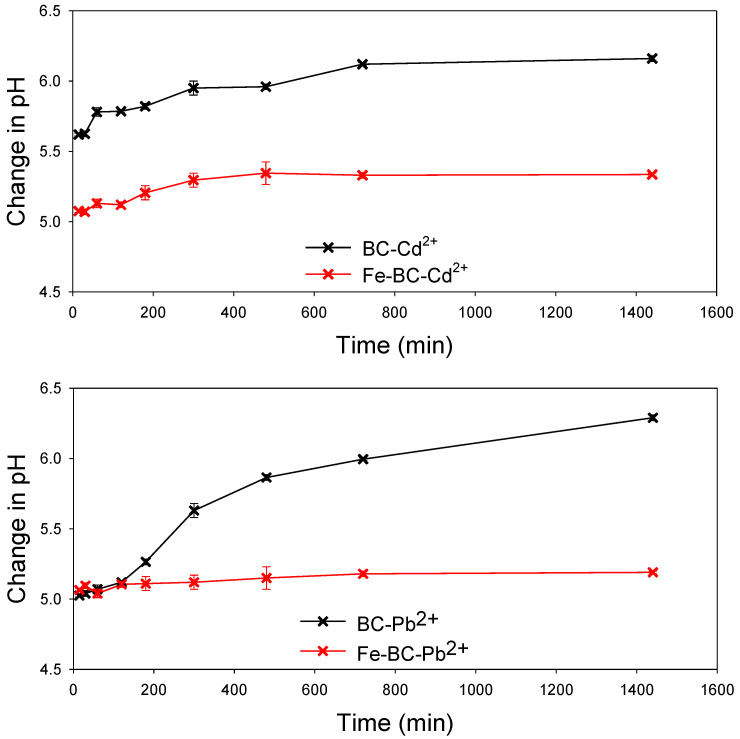
Changes in solution pH with time for the adsorption of cadmium (Cd^2+^) and lead (Pb^2+^) ions on date palm waste biochar (BC) and Fe-modified biochar (Fe-BC).

**Figure 7 ijerph-20-00802-f007:**
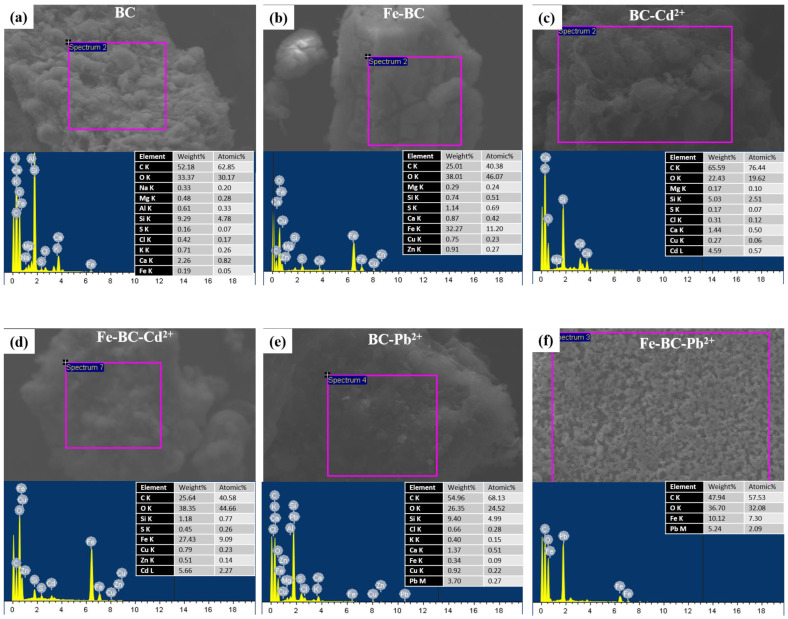
SEM–EDS analyses for (**a**) date palm waste biochar (BC), (**b**) Fe-modified biochar (Fe-BC) before the adsorption, and after the (**c**) adsorption of cadmium (Cd^2+^) onto BC, (**d**) adsorption of Cd^2+^ onto Fe-BC, (**e**) adsorption of lead (Pb^2+^) onto BC, and (**f**) adsorption of Pb^2+^ onto Fe-BC.

**Table 1 ijerph-20-00802-t001:** Some chemical and proximate analyses of date palm waste (DFS), biochar (BC), and Fe-modified biochar (Fe-BC).

	Yield (%)	Moisture (%)	Volatile Matter (%)	Resident Matter (%)	Ash (%)	pH	Cation Exchange Capacity (cmol kg^−1^)	Fe Contents (%)
DFS	-	3.21 ± 0.12	62.88 ± 2.14	18.51 ± 1.26	15.45 ± 1.08	6.78 ± 0.24	43.78 ± 2.74	-
BC	35.47 ± 3.25	1.02 ± 0.06	16.13 ± 1.87	56.79 ± 2.54	25.98 ± 2.81	9.26 ± 0.39	63.19 ± 3.52	0.12 ± 0.00
Fe-BC	-	2.08 ± 0.14	13.47 ± 1.58	33.49 ± 2.11	51.03 ± 1.25	7.12 ± 0.10	81.74 ± 3.78	19.05 ± 2.11

**Table 2 ijerph-20-00802-t002:** Elemental composition and surface characteristics of date palm waste (DFS), biochar (BC), and Fe-modified biochar (Fe-BC).

	C (%)	H (%)	N (%)	O (%)	S (%)	Surface Area (m^2^ g^−1^)	Pore Size (nm)	Total Volume in Pores (cm^3^ g^−1^)
DFS	41.54	11.87	3.08	43.45	0	3.28	31.71	0.00019
BC	68.82	6.46	1.12	24.13	0	68.49	9.18	0.0983
Fe-BC	72.98	9.36	1.23	16.77	0	119.82	4.72	0.1590

**Table 3 ijerph-20-00802-t003:** Parameters obtained from kinetic models for cadmium (Cd^2+^) and lead (Pb^2+^) adsorption on date palm waste biochar (BC) and Fe-modified biochar (Fe-BC).

		Adsorption of Cd^2+^		Adsorption of Pb^2+^	
		BC	Fe-BC	BC	Fe-BC
Pseudo-first order	*k* _1_ ^′^	−2.6 × 10^−3^	−3.0 × 10^−3^	−2.5 × 10^−3^	−3.3 × 10^−3^
	*q_e_*	2.79	3.62	3.43	4.10
	*R* ^2^	0.73	0.86	0.95	0.87
Pseudo-second order	*k* _2_ ^′^	2.9 × 10^4^	1.3 × 10^4^	4.0 × 10^−5^	4.6 × 10^−5^
	*q_e_*	27.12	54.49	45.89	105.49
	*h*	0.23	0.39	0.08	0.52
	*R* ^2^	0.97	0.98	0.93	0.95
Elovich	*a*	3.90	7.95	4.95	15.60
	*β*	−4.92	−11.10	−9.42	−25.06
	*R* ^2^	0.84	0.88	0.68	0.79
Intraparticle diffusion	*k_id_*	0.74	1.48	1.03	2.88
	*c*	2.27	3.85	1.66	4.61
	*R* ^2^	0.74	0.90	0.87	0.79
Power function	*k_f_*	0.46	0.58	0.52	0.70
	*b*	0.13	0.13	0.10	0.02
	*R* ^2^	0.98	0.97	0.85	0.95

**Table 4 ijerph-20-00802-t004:** Non-linear parameters of isotherm models for cadmium (Cd^2+^) and lead (Pb^2+^) adsorption on date palm waste biochar (BC) and Fe-modified biochar (Fe-BC).

	Sorbents	Langmuir Isotherm			Freundlich Isotherm		
		*Q_L_* (mg g^−1^)	*K_L_* (L g^−1^)	*R* ^2^	*K_F_* (L g^−1^)	1/*n*	*R* ^2^
Cd^2+^	BC	26.78	1.453	0.86	10.54	0.345	0.76
	Fe-BC	48.44	0.097	0.96	23.30	0.031	0.78
Pb^2+^	BC	160.07	0.013	0.96	36.66	0.295	0.92
	Fe-BC	475.14	0.025	0.94	129.06	0.207	0.94

**Table 5 ijerph-20-00802-t005:** Maximum adsorption capacities of different magnetic biochars for cadmium and lead from aqueous media.

Adsorbent	Modification Type	Heavy Metal	Maximum Adsorption Capacity (mg g^−1^)	Reference
BC	-	Cd^2+^	26.78	This study
		Pb^2+^	160.07	
Fe-BC	Fe^3+^/Fe^2+^	Cd^2+^	48.44	
		Pb^2+^	475.14	
Oak bark	Fe^2+^/Fe^3+^ SO_4_ solution	Cd^2+^	7.40	[37]
		Pb^2+^	30.	
Oak wood	Fe^2+^/Fe^3+^ SO_4_ solution	Cd^2+^	2.87	
		Pb^2+^	10.10	
Wheat straw	FeSO_4_7H_2_O	Cd^2+^	75.30	[38]
		Pb^2+^	311.0	
Grape husk	FeSO_4_7H_2_O	Cd^2+^	38.30	
		Pb^2+^	204	
Rice hull	Fe(acac)_3_ + calcination	Pb^2+^	23.90	[39]
Rice hull	Fe(acac)_3_ + calcination + ZnS	Pb^2+^	368.0	
Coconut shell	FeCl_3_·6H_2_O	Cd^2+^	3.846	[40]
		Pb^2+^	4.097	
Corn straw	600 °C BC + Ferric nitrate + calcination	Cd^2+^	28.71	[41]
	800 °C BC + Ferric nitrate + calcination		46.90	

## Data Availability

Not applicable.

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
