# Peer review of "Efficient Remediation of Cadmium- and Lead-Contaminated Water by Using Fe-Modified Date Palm Waste Biochar-Based Adsorbents"

_ijerph, 2023, doi:10.3390/ijerph20010802_

Round 1

Reviewer 1 Report

The work presented by Alghamdi and Alasmary shows the adsorption capacity of a biochar-modified material for both Cd (II) and Pb (II) cations. The application of various kinds of magnetic biochar for removing contaminants in water, including Cd and Pb, is discussed widely in the existent literature, as well as the preparation methods for magnetic biochar. Therefore, I would ask the authors which is the novelty of their work compared to others already published in this topic. Although the authors present a treatment of the adsorption data by adjusting to different models and analyzing the intraparticle diffusion, I do not recommend this work for publication in "International Journal of Environmental Research and Public Health", at least in the current form.

Some comments are as follows:

1. Can the authors clarify which is the novelty of their work?

2. Which is the advantage of using an adsorbent instead of the other techniques mentioned? ¿Which will be the treatment to follow once the adsorbent is exhausted?

3. The authors mention that experiments were conducted three times, but there is no statistical treatment, neither statistical bars in the graphs that show adsorption reproducibility.

4. It would be interesting to show the corresponding phase to the corresponding peaks of PXRD. In addition, which is the JCPDS or ICDS card of the different phases?

5. How was the procedure for drying of Fe-BC? Was the oxidation of Fe (II) prevented? Otherwise, which is the purpose of Fe(II)/Fe(III) mixture?

6. Which is the amount of Fe in the Fe-BC?

7. The authors show EDX images, but they do not present relevant SEM images about the morphology of the adsorbents, which are helpful for the discussion of the results. In addition, the authors present EDX analysis of this materials before and after adsorption; however, there is no information regarding the iron content in the samples.

8. There is not magnetization curve for the material, neither a proof of magnetic behaviour of the material.

9. Quality of FTIR spectra is not good enough to certify the presence of FeOOH phase. In addition, the bands characteristic of Fe-OH bonds appear in the 1000-500 cm-1 range, which is not  well resolved in this case.

10. What about the reverse process for this materials?

Regarding methodology:

1. The authors do not mention the equilibrium time at which adsorption at different concentration of metal cations is performed.

Other comments:

The text has several typos and grammatical errors. See for example:

Line 69: Numerus

Line 89: Frequently? Better to use thoroughly?

Line 96: Sieve instead screen?

Line 131: was adjusted instead adjusting

Line 135: times

Line 195: 3-unit. Moreover, the increase in pH units is not correct.

line 202: 4-fold

line 204: remove Q

Sentences of lines 76-77 is rather similar to that of lines 76-77. Merge or rewrite, since they are redundant.

Line 207: Increasing ash contents with pyrolysis temperature... but the authors only use one temperature for pyrolysis! Why this sentence?

Author Response

Reviewer 1

Comment 1: Can the authors clarify which is the novelty of their work?

Response: We are highly thankful to the reviewer for valuable comments and suggestions, which were very helpful in improving the quality of our manuscript. We have considered all the comments/suggestions and revised the manuscript thoroughly. The novelty statement for this study has been added in last paragraph of the introduction section “Therefore, magnetic-BC composites are more conducive and efficient for contaminant removal, however, most of the methods to synthesize magnetic-BC are complicated. Moreover, there is possibility of magnetic particles to occupy the adsorption sites of BC. Thus, new techniques are needed to design highly efficient magnetic-BC for heavy metals removal from the environment”.

Comment 2: Which is the advantage of using an adsorbent instead of the other techniques mentioned? Which will be the treatment to follow once the adsorbent is exhausted?

Response: Adsorption is extensively regarded as a promising technique for the removal of toxic compounds from the environment. The advantages of adsorption include simplicity, high efficiency, re-usability, environment-friendly, and low cost. In case of biochar, it is prepared from waste materials through thermal treatment process, which is one of the cheapest way to fabricate efficient adsorbents. Generally, the adsorbents can be reused by regeneration process after they exhaust.

Comment 3: The authors mention that experiments were conducted three times, but there is no statistical treatment, neither statistical bars in the graphs that show adsorption reproducibility.

Response: All the adsorption trials were conducted in 3 replications. We have added the error bars and standard deviations in the data where necessary (Table 1, Fig. 5, and Fig. 6). However, the adsorption models including isotherm and kinetics models were applied to the average values, and thus, error bars were not included. Moreover, all the modeling was performed with SigmaPlot, which does not include error bars for the isotherms calculations.

Comment 4:  It would be interesting to show the corresponding phase to the corresponding peaks of PXRD. In addition, which is the JCPDS or ICDS card of the different phases?

Response: Thank you for your suggestion. We have provided the PCPDs for magnetic peaks i.e., “Same happened with calcite mineral (29.21° 2θ: JCPDS: 47-1743), which was substantially reduced in BC and Fe-BC as compared to DFS. Akin to FTIR, the presence of the Fe particles was also witnessed by XRD results of Fe-BC by observing a peak at 44.21 and 63.25° 2θ (JCPDS 00-001-1267). Likewise, a peak representing hematite (JCPDS Card No. 01-073-0603) was found at 35.54° 2θ in Fe-BC material”.

Comment 5: How was the procedure for drying of Fe-BC? Was the oxidation of Fe (II) prevented? Otherwise, which is the purpose of Fe(II)/Fe(III) mixture?

Response: Thank you for highlighting this point. The Fe-BC composites were dried in vacuum oven to avoid oxidation. It has been updated in the revised manuscript.

Comment 6: Which is the amount of Fe in the Fe-BC?

Response: Thank you for this comment. The contents of Fe in BC were 0.12%, while in Fe-BC were 19.05%. This information has been added in Table 1 in the revised manuscript.

Comment 7:  The authors show EDX images, but they do not present relevant SEM images about the morphology of the adsorbents, which are helpful for the discussion of the results. In addition, the authors present EDX analysis of this materials before and after adsorption; however, there is no information regarding the iron content in the samples.

Response: The EDX images have been revised and modified according to your suggestion. The contents of iron have been included in the image now. The images of SEM are also in the EDX analyses.

Comment 8: There is not magnetization curve for the material, neither a proof of magnetic behaviour of the material.

Response: We are agreed with your comment. However, the objectives of this study were to design an efficient adsorbent for Cd and Pb removal from contaminated water. We have modified the BC with iron to prepare efficient magnetic adsorbent, but we provided basic characterization of the material and could not investigate its magnetic characteristics.

Comment 9: Quality of FTIR spectra is not good enough to certify the presence of FeOOH phase. In addition, the bands characteristic of Fe-OH bonds appear in the 1000-500 cm-1 range, which is not well resolved in this case.

Response: Thank you for pointing out it. We have improved the quality of the FTIR graphs and added some new description for FeOOH in Fe-BC material.

Comment 10:  What about the reverse process for this materials?

Response: The used adsorbents were tested for regeneration for up to 4 cycles. Regeneration studies showed that Cd2+ and Pb2+ removal efficiencies of Fe-BC were 29.58% and 45.31% after 4 cycles, respectively, which was lower than that of BC (38.13% and 53.38%, respectively). Reduction in Fe-BC efficiency could be due to oxidation and dissociation of Fe particles.

Comment 11:  The authors do not mention the equilibrium time at which adsorption at different concentration of metal cations is performed.

Response: The equilibrium studies were conducted for 1440 min of equilibrium time. This is added into the revised manuscript.

Comment 12: The text has several typos and grammatical errors. See for example:

Response: Thank you for mentioning these mistakes. We have revised the manuscript thoroughly and removed all these mistakes.

  1. Line 69: Numerus

Response: It is corrected as “numerous”

  1. Line 89: Frequently? Better to use thoroughly?

Response: It is corrected.

  • Line 96: Sieve instead screen?

Response: “Screen” is replaced with “sieve”

  1. Line 131: was adjusted instead adjusting

Response: It is corrected now.

  1. Line 135: times

Response: It is corrected.

  1. Line 195: 3-unit. Moreover, the increase in pH units is not correct.

Response: It is corrected now.

  • line 202: 4-fold

Response: It is corrected.

  • line 204: remove Q

Response: Q id removed

  1. Sentences of lines 76-77 is rather similar to that of lines 76-77. Merge or rewrite, since they are redundant.

Response: It is corrected.

  1. Line 207: Increasing ash contents with pyrolysis temperature... but the authors only use one temperature for pyrolysis! Why this sentence?

Response: This statement has been modified in the revised manuscript.

Reviewer 2 Report

Comments for Author, 

I have studied this work thoroughly and in detail. Overall, the work is well-planned and novelty. Generally, the work is well-planned and original. However, the academic English of the study is sufficient, and minor grammatical adjustments are required.

The abstract part needs to be detailed, and at the same time, a few statements about pollution need to be included.

Introduction:

Please support these phrases with refs " The demand for fresh and clean water resources is increasing rapidly due to prompt 28 increase in global population"
This sentence is unclear. Please correct it. 
"Recently, BC modification with any foreign material has gained much attention as it 76 improves the physiochemical characteristic of BC for targeted applications"

Result and Discussion:
If possible, add lettering to your table and figures. Most of the data analyses are unclear. 

The author mentioned many places about pollution sources, but most of the parts are properly referenced. 

I think the used methods are selected carefully and The discussion is needed to support with more detail please discuss more with statistical results. The author mentioned many places regarding other biochar applications

A conclusion is needed to extend future recommendations. 

Best regards

Author Response

Reviewer 2

Comment 1: I have studied this work thoroughly and in detail. Overall, the work is well-planned and novelty. Generally, the work is well-planned and original. However, the academic English of the study is sufficient, and minor grammatical adjustments are required.

Response: Thank you for your time to review our manuscript. We have considered all of the suggestions and comments, and it really helped us improving the quality of our manuscript. We have revised the manuscript thoroughly and rectified all the mistakes.

Comment 2: The abstract part needs to be detailed, and at the same time, a few statements about pollution need to be included.

Response: The abstract has been revised and improved as per your suggestion.

Comment 3: Please support these phrases with refs " The demand for fresh and clean water resources is increasing rapidly due to prompt increase in global population".

Response: The reference is added for this statement.

Comment 4: This sentence is unclear. Please correct it. "Recently, BC modification with any foreign material has gained much attention as it 76 improves the physiochemical characteristic of BC for targeted applications"

Response: This sentence has been revised and corrected.

Comment 5: If possible, add lettering to your table and figures. Most of the data analyses are unclear. 

Response: Thank you for this suggestion. The statistical analyses data has been performed and standard deviation as well as error bars have been added where necessary (Table 1, Figure 5 and Figure 6). However, most of the Tables and Figures are generated through isotherm and kinetics modeling, where average values were used during the modeling process.

Comment 6: The author mentioned many places about pollution sources, but most of the parts are properly referenced. 

Response: Thank you for this encouraging comment. We have improved the quality of the manuscript further.

Comment 7: I think the used methods are selected carefully and The discussion is needed to support with more detail please discuss more with statistical results. The author mentioned many places regarding other biochar applications.

Response: The discussion section has been improved now. We have added the standard deviations in the data. The data for regeneration studies were also added into the revised manuscript.

Comment 8: A conclusion is needed to extend future recommendations. 

Response: The conclusion section has been revised and improved thoroughly.

Reviewer 3 Report

Biochar material has been obtained from natural source, and modified by in situ synthesis of iron oxide nanoparticles. The such obtained materials were applied for sorptive removal of heavy metal ions from their aqueous solutions. The results showed that iron-modified material had a substantially higher sorption capacity to both metal ions. The motivation of the study is appropriate, and the obtained results are potentially useful for development of economical and environment-friendly materials and technologies for polluted water remediation.

The work seems to be acceptable with minor revisions.

1. Figure 4 is somewhat confuse. The experimental data should be named as data, and the isotherm lines should have their appropriate names.

2. In Figure 4 b the red line does not seem to be a Frendlich isotherm, since it does not increase toward high Ce. Authors are recommended to check the used models and fits.

3. in the SEM/EDS figure, the scale bar should be added well visibly.

4. in the EDS figure, the analysis should probably include Fe, otherwise the calculated compositions will be erroneous.

5. The performance of the material towards binding these two metal ions should be compared to other magnetically modified sorbent materials.

6. in References, the journal names and abbreviations should be checked and brought to the mdpi style.

7. Thorough spell check is needed for the paper.

Author Response

Reviewer 3

Comment: Biochar material has been obtained from natural source, and modified by in situ synthesis of iron oxide nanoparticles. The such obtained materials were applied for sorptive removal of heavy metal ions from their aqueous solutions. The results showed that iron-modified material had a substantially higher sorption capacity to both metal ions. The motivation of the study is appropriate, and the obtained results are potentially useful for development of economical and environment-friendly materials and technologies for polluted water remediation. The work seems to be acceptable with minor revisions.

Response: We are highly thankful to the reviewer for valuable comments and suggestions. We have considered all the comments/suggestion and modified our manuscript accordingly. The comments/suggestions were helpful in improving the quality of our manuscript.

Comment 1: Figure 4 is somewhat confused. The experimental data should be named as data, and the isotherm lines should have their appropriate names.

Response: Thank you for your comment. Figure 4 explains the isotherm models. We have employed 2 isotherm models i.e., Freundlich and Langmuir isotherms. In Figure 4, “4a” is for Freundlich isotherm for Pb sorption, 4b is Freundlich isotherm for Cd sorption, 4c is Langmuir isotherm for Pb sorption, and 4d is Langmuir isotherm for Cd sorption. Actually, we have tried many ways to improve the visibility and clarity of the figure, but this is the most suitable way where the figure is more understandable and clear.

Comment 2: In Figure 4 b the red line does not seem to be a Frendlich isotherm, since it does not increase toward high Ce. Authors are recommended to check the used models and fits.

 Response: Thank you for pointing out this. We have confirmed Figure 4b, and it is actually Freundlich isotherm. It increases with higher Ce value but up to a certain limit and then acquires equilibrium stage. However, this red line is for BC, and to compare it with Fe-BC, we have increased the scale of the graphs, which could be reason that it looks not increasing towards higher Ce.

Comment 3: in the SEM/EDS figure, the scale bar should be added well visibly.

 Response: We are agreed with your suggestion. However, unfortunately, we do not have access to the data files as we got analyzed these samples from another organization and we do not have this facility in our department. We were sent these figures of SEM-EDS and we are unable to edit them. However, we have revised these and improved the quality.

Comment 4: in the EDS figure, the analysis should probably include Fe, otherwise the calculated compositions will be erroneous.

Response: We have revised the EDS figures and added elemental composition with Fe contents in the revised manuscript.

Comment 5: The performance of the material towards binding these two metal ions should be compared to other magnetically modified sorbent materials.

 Response: Thank you for your suggestion. We have compared the adsorption capacities of different magnetic biochars. The data has been presented in Table 5, and also described in the results and discussion section.  

Comment 6: in References, the journal names and abbreviations should be checked and brought to the mdpi style.

 Response: Thanks for this suggestion. The abbreviations have been checked and modified where necessary.

Comment 7: Thorough spell check is needed for the paper.

Response: The spell and English language has been checked and thoroughly and the mistakes have been rectified.

Reviewer 4 Report

1. Overview and general recommendation:

I enjoyed reviewing the manuscript of “Efficient Remediation of Cadmium and Lead-Contaminated Water by Using Fe-modified Date Palm Waste Biochar Based Adsorbents”. Water pollution has long been a problem, and heavy metals, as one of the hazardous contaminants, have raised great attention. Biochar (BC) is widely used for pollution treatment due to its unique properties, though it has a few drawbacks. Therefore, it is necessary to make modifications to achieve higher efficiency.

In this study, the authors investigated removal of Cd2+ and Pb2+ using BC and magnetic biochar (Fe-BC) through adsorption experiments coupled with a series of characterizations. They found that Fe-BC showed better performance in removing Cd2+ and Pb2+, and both BC and Fe-BC showed higher removal for Pb2+ than Cd2+.

Overall, the paper is well written and straightforward. However, there are a few things that still need to be addressed. Therefore, I recommend that minor revision is warranted.

2. Detailed comments:

General comments: I suggest you conduct a more thorough literature review and make a sound summary to highlight the novelty of this work. Now the research gap is not clear to readers.

Detailed comments are listed below.

(1) Page 1, Line 16. “than” not “Than”.

(2) Pages 2, Line 63. “Easy” rather than “easiness”.

(3) Page 2, Lines 74-75. Add more references regarding the current progress of such modification and briefly introduce some key findings. What has been done by other researchers, and what are the new insights this study can bring?

(4) Page 2, Line 78. Should be “it is also challenging to separate BC particles from the liquid-solid phase due to their small size”.

(5) Page 2, Line 88. “waste” not “wastes”.

(6) Page 2, Lines 93-94. How did you ensure anaerobic conditions? Please provide some details.

(7) Page 3, Line 102. Please provide the vendor information of the chemicals used.

(8) Page 3, Line 113. Delete “of”. Please check carefully throughout the paper for similar mistakes and correct them.

(9) Page 3, Line 127. Replace “specific amount” with the exact weight.

(10) Page 3, Line 135. Similarly, specify the time instead of just saying “specific period of time”.

(11) Page 4, Lines 148-152. What are the criteria of selecting heavy metal concentrations and pH used, and their relation to reality? I know you discussed the pH in Results and Discussion section, but you should also mention it in Experimental section.

(12) Page 5, Line 217. Please provide the full name of BET.

(13) Page 5, Line 227. Is this statement just your hypothesis or you can actually find some references to cite and support it?

(14) Page 12, Lines 381-390. Change all the Pb+2/Cd+2 to Pb2+/Cd2+.

Author Response

Reviewer 4

Overview and general recommendations:

Comment: In this study, the authors investigated removal of Cd2+ and Pb2+ using BC and magnetic biochar (Fe-BC) through adsorption experiments coupled with a series of characterizations. They found that Fe-BC showed better performance in removing Cd2+ and Pb2+, and both BC and Fe-BC showed higher removal for Pb2+ than Cd2+. Overall, the paper is well written and straightforward. However, there are a few things that still need to be addressed. Therefore, I recommend that minor revision is warranted.

Response: We are highly thankful to the reviewer for valuable comments and suggestions which were very helpful in improving the quality of our manuscript. We have considered all the comments/suggestions and modified the manuscript accordingly. The revised version of the manuscript is in a good shape now.

General comments

Comment: General comments: I suggest you conduct a more thorough literature review and make a sound summary to highlight the novelty of this work. Now the research gap is not clear to readers.

Response: Thank you for this very important suggestion. We have revised the manuscript thoroughly and added a novelty statement in the last paragraph of the introduction section as “Recently, modification of BC with foreign material via physiochemical treatments has gained much attention as it improves the physiochemical characteristic of BC and increase it efficacy for targeted applications [19]. Moreover, the small size of the BC particles is also challenging to separate it from the liquid-solid phase. Therefore, magnetic-BC composites are more conducive and efficient for contaminant removal, however, most of the methods to synthesize magnetic-BC are complicated. Moreover, there is possibility of magnetic particles to occupy the adsorption sites of BC. Thus, new techniques are needed to design highly efficient magnetic-BC for heavy metals removal from the environment”.

Detailed comments

Comment 1: Page 1, Line 16. “than” not “Than”.

Response: Thank you for indicating this mistake. It has been rectified in the revised manuscript.

Comment 2: Pages 2, Line 63. “Easy” rather than “easiness”.

Response: The word “easiness” has been replaced with “easy”

Comment 3: Page 2, Lines 74-75. Add more references regarding the current progress of such modification and briefly introduce some key findings. What has been done by other researchers, and what are the new insights this study can bring?

Response: Thank you for this suggestion. We have added more references along with some recent findings. Overall, we have improved the manuscript and updated the references throughout the manuscript.

Comment 4: Page 2, Line 78. Should be “it is also challenging to separate BC particles from the liquid-solid phase due to their small size”.

Response: The text is modified as “Moreover, it is also challenging to separate BC particles from the liquid-solid phase due to their small size”.

Comment 5: Page 2, Line 88. “waste” not “wastes”.

Response: “Wastes” is replaced with “waste”

Comment 6: Page 2, Lines 93-94. How did you ensure anaerobic conditions? Please provide some details.

Response: Thank you for highlighting this mistake. In fact, we have not kept anaerobic condition, rather, we used limited oxygen conditions. We have used stainless steel closed containers. Although the containers were tightly closed, but those were not air-tight, and a limited oxygen can enter the containers. We have modified this text in the revised manuscript.   

Comment 7: Page 3, Line 102. Please provide the vendor information of the chemicals used.

Response: These information has been added into the revised version of the manuscript.

Comment 8: Page 3, Line 113. Delete “of”. Please check carefully throughout the paper for similar mistakes and correct them.

Response: Thank you for this suggestion. We have thoroughly revised the manuscript and rectified all English language and grammatical mistakes.

Comment 9: Page 3, Line 127. Replace “specific amount” with the exact weight.

Response: “specific amount” has been replaced with “the exact weight”

Comment 10: Page 3, Line 135. Similarly, specify the time instead of just saying “specific period of time”.

Response: The specific time range has been provided here, however, all specific time details are given in the next paragraph i.e., 0, 15, 30, 60, 120, 180, 300, 480, and 720, and 1440 min.

Comment 11: Page 4, Lines 148-152. What are the criteria of selecting heavy metal concentrations and pH used, and their relation to reality? I know you discussed the pH in Results and Discussion section, but you should also mention it in Experimental section.

Response: Thank you for this suggestion. At pH 5, the Cd2+ or Pb2+ ions are free and mobile in aqueous solution. The used heavy metals concentrations are the maximum concentration of Cd2+ or Pb2+ that can be found in the wastewater.

Comment 12: Page 5, Line 217. Please provide the full name of BET.

Response: Thank you for pointing out this mistake. We have added the full name of BET i.e., Brunauer-Emmett-Teller.

Comment 13: Page 5, Line 227. Is this statement just your hypothesis or you can actually find some references to cite and support it?

Response: The relevant references have been added to support this statement.

Comment 14: Page 12, Lines 381-390. Change all the Pb+2/Cd+2 to Pb2+/Cd2+.

Response: The terms “Pb+2/Cd+2” have been replaced with “Pb2+/Cd2+” throughout the manuscript.

Round 2

Reviewer 1 Report

The authors have provided a revised version of the manuscript. Now the work has been improved; however, there are some points, to my opinion, that are not clear.

To know: 

I understand that magnetic-modified biochar can be a better material for adsorption due to the simplified separation from the liquid medium once finished the adsorption. This is one of the novelties that the authors mention about this work; however, there is no proofs about this property in the materials presented. Since it is one of the most relevant aspects in this work it cannot be considered as 'out-of-scope' as the authors answer. I think that this property should be proven in a way or another.

I could not find standard deviation in Table 1 as the authors mention in response 3.

Regarding Response 6 and the data associated to it, it can be seen that Fe content in the modified material is 19.05%. After adsorption experiments, it decreases to 9 and 7%. This suggest loss of supported Fe by leaching during the tests. This could be one of the reasons for the drop of activity. 

By the way, the authors do not mention the whole protocol for reusing of material. This is washed with HCl... how long? Which concentration? How this affect to the Fe content?

There are still several mistakes in the text, for instance in line 80 it should be increases; line 155--> times; line 226 --> fold; line 228 --> remove Q

Sentence of lines 222-223 should be revised

Sentence in lines 231-232 should be revised

Author Response

Reviewer 1

Comment 1: I understand that magnetic-modified biochar can be a better material for adsorption due to the simplified separation from the liquid medium once finished the adsorption. This is one of the novelties that the authors mention about this work; however, there is no proofs about this property in the materials presented. Since it is one of the most relevant aspects in this work it cannot be considered as 'out-of-scope' as the authors answer. I think that this property should be proven in a way or another.

Response: We are highly thankful to the reviewer for valuable comments and suggestions, which were very helpful in improving the quality of our manuscript. We have considered all the comments/suggestions and revised the manuscript thoroughly. Regarding your suggestion about magnetic properties of Fe-BC, unfortunately, we could not conduct magnetic hysteresis loops. However, the recovery of magnetic biochar has been shown in Figure 5 in the revised version, and the magnetic recovery of Fe-BC has been explained in the discussion section as “The reduction in Fe contents from 32.27% in Fe-BC to 27.43% in Cd2+-loaded Fe-BC and 10.12% in Pb2+-loaded Fe-BC suggested the leaching of Fe-particles during the adsorption trials, which might has resulted in reducing the adsorption capacity of these adsorbents after 4 cycles of regeneration as compared to BC. On contrary, Fe-BC showed higher recovery under applied magnetic field as compared to BC (Figure 5), which suggests the higher potential of Fe-BC for regeneration and repeated application. However, even with the reduced activity of Fe-BC after 4 regenerations, its Pb2+ and Cd2+ removal efficiency was relatively higher than most of the previously reported magnetic biochars”.

Comment 2: I could not find standard deviation in Table 1 as the authors mention in response 3.

Response: Thank for pointing out. Actually, we added the standard deviations in the Table, but forgot to replace the table before submitting the revised manuscript. However, the new table with standard deviation has been added into the manuscript now.

Comment 3: Regarding Response 6 and the data associated to it, it can be seen that Fe content in the modified material is 19.05%. After adsorption experiments, it decreases to 9 and 7%. This suggest loss of supported Fe by leaching during the tests. This could be one of the reasons for the drop of activity. 

Response: Yes, we are agreed with your suggestion. Due to oxidation and dissociation of Fe particles, followed by leaching could have resulted in reduced activity of Fe-BC as compared to BC. It was obvious in regeneration studies as well. This explanation has been added in the revised version of the manuscript as “The reduction in Fe contents from 32.27% in Fe-BC to 27.43% in Cd2+-loaded Fe-BC and 10.12% in Pb2+-loaded Fe-BC suggested the leaching of Fe-particles during the adsorption trials, which might has resulted in reducing the adsorption capacity of these adsorbents after 4 cycles of regeneration as compared to BC”.

Comment 4:  By the way, the authors do not mention the whole protocol for reusing of material. This is washed with HCl... how long? Which concentration? How this affect to the Fe content?

Response: The regeneration protocol has been updated with more detailed information now. The new text is as “The regeneration trials of Fe-BC and BC were conducted using 0.2 M HCl up to 4 cycles. The initial concentrations for Cd2+ and Pb2+ in regeneration studies were 100 and 600 mg L-1, respectively. Briefly, Cd2+ or Pb2+–loaded adsorbents were washed with deionized water to remove surplus metal, and dried in oven at 60°C. Thereafter, the dried adsorbents loaded with Cd2+ or Pb2+ were suspended into 30 mL 0.2 M HCl solution and shaken for 1440 min at 150 rpm. Then, the solutions were separated from the adsorbent and subjected to Cd2+ and Pb2+ analyses by ICP-OES, while the adsorbents were washed with deionized water, dried, and subjected to another cycle of adsorption-desorption”.   

Comment 5: There are still several mistakes in the text, for instance in line 80 it should be increases; line 155--> times; line 226 --> fold; line 228 --> remove Q

Response: Thank your indicating these mistakes.

  • “increase” has been replaced with “increases”
  • “time” is changed to “times”
  • “folds” is replaced with “fold”
  • “Q” has been removed from the sentence.

Comment 6: Sentence of lines 222-223 should be revised

Response: Thank you for highlighting this mistake. This sentence has been revised now.

Comment 7:  Sentence in lines 231-232 should be revised

Response: This sentence has been revised and modified in the revised version of the manuscript.

Reviewer 2 Report

I have read your revised paper carefully, The author improved paper and the paper could be accepted for the publication.

Best Regards

Author Response

Reviewer 2

I have read your revised paper carefully, the author improved paper and the paper could be accepted for the publication.

Response: Thank you for sparing your valuable time to review this manuscript. We really appreciate your efforts to improve the quality of our manuscript.